# GluN2B-containing NMDA receptors regulate depression-like behavior and are critical for the rapid antidepressant actions of ketamine

**Oliver H Miller[1†], Lingling Yang[2†], Chih-Chieh Wang[1], Elizabeth A Hargroder[1], Yihui Zhang[2], Eric Delpire[3], Benjamin J Hall[1,2,4]***

[1]Neuroscience Program, Tulane University, New Orleans, United States; [2]Department of Cell and Molecular Biology, Tulane University, New Orleans, United States; [3]Department of Anesthesiology, Vanderbilt University Medical Center, Nashville, United States; [4]Roche Pharmaceutical Research and Early Development, Neuroscience, Ophthalmology, and Rare Diseases, Roche Innovation Center, Basel, Switzerland

**Abstract** A single, low dose of the NMDA receptor antagonist ketamine produces rapid antidepressant actions in treatment-resistant depressed patients. Understanding the cellular mechanisms underlying this will lead to new therapies for treating major depression. NMDARs are heteromultimeric complexes formed through association of two GluN1 and two GluN2 subunits. We show that in vivo deletion of GluN2B, only from principal cortical neurons, mimics and occludes ketamine's actions on depression-like behavior and excitatory synaptic transmission. Furthermore, ketamine-induced increases in mTOR activation and synaptic protein synthesis were mimicked and occluded in 2BΔCtx mice. We show here that cortical GluN2B-containing NMDARs are uniquely activated by ambient glutamate to regulate levels of excitatory synaptic transmission. Together these data predict a novel cellular mechanism that explains ketamine's rapid antidepressant actions. In this model, basal glutamatergic neurotransmission sensed by cortical GluN2B-containing NMDARs regulates excitatory synaptic strength in PFC determining basal levels of depression-like behavior.

*For correspondence: benhall@tulane.edu

†These authors contributed equally to this work

**Competing interests:** The authors declare that no competing interests exist.

## Introduction

A single, sub-anesthetic dose of the n-methyl d-aspartate (NMDA) receptor antagonist ketamine produces anti-depressant effects in treatment-resistant depressed patients (*Berman et al., 2000*; *Zarate et al., 2006*). Understanding the cellular signaling mechanisms underlying this effect will lead to new therapeutic strategies, with the potential for enhancing beneficial actions and minimizing side effects associated with treatment regimes. Ketamine evokes a rapid increase in protein synthesis and enhances excitatory synaptic transmission in cortical neurons (*Li et al., 2010*; *Autry et al., 2011*). However, it's unclear how *suppression* of NMDA receptor (NMDAR) signaling *promotes* protein synthesis.

The cortical NMDAR complex is heteromultimeric, containing two GluN1 and two GluN2 subunits, the latter of which are encoded by four genes (GluN2A-D) (*Monyer et al., 1992*). Cortical NMDARs are dominated by GluN2A and GluN2B subunits. We recently demonstrated that GluN2B-containing NMDARs act in a unique manner, distinct from GluN2A, to directly suppress mammalian target of rapamycin (mTOR) signaling and repress protein synthesis (*Wang et al., 2011a*). Consistent with a role for GluN2B, selective antagonists of GluN2B-containing NMDARs are effective in producing rapid

**eLife digest** Depression is the leading cause of disability worldwide, with hundreds of millions of people living with the condition. The 'gold standard' for depression treatment involves a combination of psychotherapy and medication. Unfortunately, current antidepressant medications do not help everyone, waiting lists for psychotherapy are often long, and both normally take a number of weeks of regular treatment before they begin to have an effect. As patients are often at a high risk of suicide, it is crucial that treatments that act more quickly, and that are safe and effective, are developed.

One substance that may fulfill these requirements is a drug called ketamine. Studies have shown that depression symptoms can be reduced within hours by a single low dose of ketamine, and this effect on mood can last for more than a week. However, progress has been hindered by a lack of knowledge about what ketamine actually does inside the brain.

Neurons communicate with one another by releasing chemicals known as neurotransmitters, which transfer information by binding to receptor proteins on the surface of other neurons. Drugs such as ketamine also bind to these receptors. Ketamine works by blocking a specific receptor called the n-methyl d-aspartate (NMDA) receptor, but how this produces antidepressant effects is not fully understood.

The NMDA receptor is actually formed from a combination of individual protein subunits, including one called GluN2B. Now Miller, Yang et al. have created mice that lack receptors containing these GluN2B subunits in neurons in their neocortex, including the prefrontal cortex, a brain region involved in complex mental processes such as decision-making. This allowed Miller, Yang et al. to discover that when the neurotransmitter glutamate binds to GluN2B-containing NMDA receptors, it limits the production of certain proteins that make it easier for signals to be transmitted between neurons. Suppressing the synthesis of these proteins too much may cause depressive effects by reducing communication between the neurons in the prefrontal cortex.

Both mice lacking GluN2B-containing receptors in their cortical neurons and normal mice treated with ketamine showed a reduced amount of depressive-like behavior. This evidence supports Miller, Yang et al.'s theory that by blocking these NMDA receptors, ketamine restricts their activation. This restores normal levels of protein synthesis, improves communication between neurons in the cortex, and reduces depression.

Understanding how ketamine works to alleviate depression is an important step towards developing it into a safe and effective treatment. Further research is also required to determine the conditions that cause overactivation of the GluN2B-containing NMDA receptors.

changes in behavior in both clinical patient populations and rodent models of depression (*Li et al., 2010*) (*Maeng et al., 2008*; *Preskorn et al., 2008*; *Li et al., 2011*). However, it is unknown how *selective* antagonism of GluN2B-containing receptors produces similar effects as antagonizing NMDARs using *non-subunit selective* antagonists.

We hypothesized that ambient glutamate tonically activates GluN2B-containing NMDARs to basally, and directly, suppress protein synthesis in principal cortical neurons and that antagonism of this action, either by GluN2B-selective or pan-NMDAR antagonists, would initiate the rapid antidepressant effects by increasing protein synthesis and enhancing excitatory synaptic transmission in prefrontal cortex (PFC). This hypothesis predicts that genetic deletion of GluN2B selectively from principal cortical neurons should mimic and occlude the actions of ketamine on depression-like behaviors and excitatory synaptic transmission. To test this, we generated animals with selective genetic knockout of GluN2B in principal cortical neurons (2BΔCtx) by crossing mice with a conditional GluN2B KO allele (*Brigman et al., 2010*) and mice expressing Cre-recombinase (Cre) under control of the NEX promoter (*Goebbels et al., 2006*). We then sequentially measured behavior, excitatory cortical synapse physiology, and synaptic protein expression following single dose ketamine injection compared to saline-injected control animals.

We show here that genetic deletion of GluN2B from principal cortical neurons both mimics and occludes the effects of ketamine in suppression of depression-like behavior and increased frequency of individual excitatory synaptic events onto layer II/III pyramidal neurons in PFC. We also show that

mTOR is present in synaptic protein fractions of cortical lysates and ketamine induces a rapid, yet transient, increase in mTOR phosphorylation, which is occluded in 2BΔCtx animals. Cortical GluN2B removal also eliminated susceptibility to chronic corticosterone exposure. Furthermore, GluN2B-containing receptors can be uniquely activated by ambient glutamate, supporting a model whereby GluN2B maintains tonic suppression of protein synthesis in principal cortical neurons. In support of this, we show that modulation of glutamate transporter function, in vivo, bidirectionally regulates excitatory synaptic transmission and that enhancing glutamate transporter function suppresses depression-like behavior while increasing excitatory synaptic drive in PFC. In summary, our data suggest a novel mechanistic model for the antidepressant actions of ketamine that involves tonic activation of GluN2B-containing NMDARs in helping set basal levels of despair through regulation of protein synthesis and excitatory synaptic drive in PFC.

## Results

### Removal of GluN2B from principal cortical neurons: 2BΔCtx

To test the importance of cortical GluN2B-containing NMDARs in regulating despair-like behavior and excitatory synaptic transmission, we generated cortex- and principal neuron-specific GluN2B knockout animals (2BΔCtx) by crossing mice carrying a Lox-P flanked GluN2B allele (*Brigman et al., 2010*) with animals containing a Cre-recombinase (Cre) cassette expressed in principal neurons of the neocortex: NEXCre (*Goebbels et al., 2006*) (*Figure 1*). We first confirmed this genetic technique resulted in the removal of GluN2B protein by PCR and western blot analyses. PCR analysis of genomic DNA isolated from tail tissue confirmed the presence of both the NEXCre and GluN2B-floxed alleles in 2BΔCtx mice (*Figure 1A*). For all experiments involving 2BΔCtx mice, experimental animals (NEX$^{Cre/+}$ : GluN2B$^{flox/flox}$) were compared to littermate controls (either NEX$^{+/+}$ : GluN2B$^{flox/flox}$ or NEX$^{+/+}$ : GluN2B$^{flox/+}$). In contrast to brainstem lysates, cortical lysates from 2BΔCtx animals at P10 showed significant decrease in GluN2B expression compared to protein samples from controls (*Figure 1B*). GluN2B protein levels were also significantly reduced at P50–P70 and were not accompanied by any statistically significant change in expression of either GluN1 or GluN2A (*Figure 1B*). Residual GluN2B protein is due to the expression in non-principal neurons including inhibitory interneurons.

As expected, crossing NEX-cre mice with a dsRed flox-stop-flox GFP reporter line (JAX stock #008705) resulted in strong GFP signal in cortical brain regions while subcortical, cerebellar, and brain stem structures expressed RFP due to the absence of cre expression in these structures. In PFC, 80.9 ± 1.3% of neurons were GFP positive. We also noted that cre expressed strongly in both ventral and dorsal aspects of the hippocampus, in contrast to the cortex-restricted CaMKII promoter-driven cre mouse line, which lacks expression in the ventral hippocampus (*Figure 1C*) (*Tsien et al., 1996*; *Brigman et al., 2010*). Consistent with previous reports characterizing this cre encoding animal (*Goebbels et al., 2006*), we observed GFP expression in these reporter mice in pyramidal neurons in the neocortex. To confirm that this genetic manipulation resulted in loss of GluN2B-containing NMDAR-mediated current at cortical synapses, we applied whole-cell voltage-clamp recordings in acute brain slices. Intra-columnar electrical stimulation in slices from PFC of control animals evoked strong synaptic currents in layer II/III pyramidal neurons (*Figure 1D,E*). By voltage clamping at −65 mV, to maximize the Mg$^{2+}$ block of NMDARs and thereby isolate AMPAR-mediated current, we observed rapid onset, inward synaptic current in both genotypes confirming the presence of functional AMPAR-containing excitatory synapses. At depolarized potentials (+50 mV) these evoked synaptic responses are a mix of non-rectifying AMPAR-mediated current and slower decaying NMDAR-mediated current. Measuring the relative current carried by AMPARs to NMDARs, we observed a significant increase in this ratio in 2BΔCtx animals, indicating either a decrease in NMDAR-mediated signaling, an increase in AMPAR-mediated signaling, or both (*Figure 1E*). We did not observe any differences in the decay kinetics of these AMPAR-mediated EPSCs (tau for Cont = 10.2 ± 0.4 ms vs 2BΔCtx = 11.4 ± 3.4 ms, n.s.) nor in integrated charge transfer at −70 mV (Cont = 1.33 ± 0.67 pC vs 2BΔCtx = 1.77 ± 0.53 pC, n.s.). Importantly, as predicted by the loss of GluN2B expression, sensitivity of the NMDAR-mediated current recorded at +50 mV to the GluN2B-selective antagonist ifenprodil (3 µM), which was approximately 40% in control slices, was completely lost in 2BΔCtx animals (*Figure 1D,E*). Furthermore, loss in ifenprodil sensitivity was accompanied by a significant decrease in the decay tau of the receptor complex, consistent with the fact that GluN2A-containing NMDARs undergo more rapid receptor deactivation (*Flint et al., 1997*). Together with the genetic and biochemical data, these

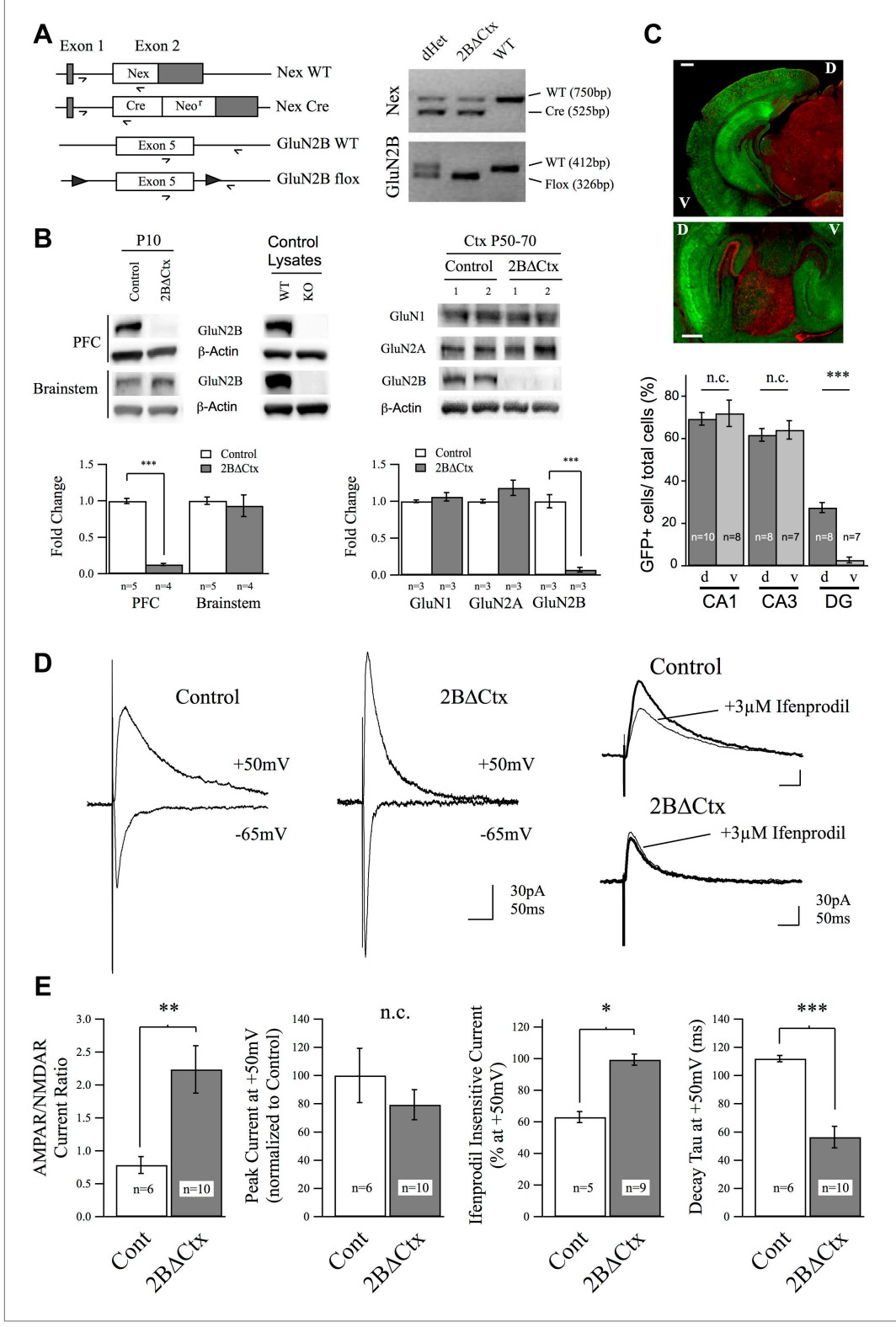

**Figure 1**. Genetic knockout of GluN2B from principal cortical neurons in vivo. (**A**) Conditional 'floxed' GluN2B knockout mice were crossed with NEX-Cre animals to ablate GluN2B from principal cortical neurons in vivo (2BΔCtx). Genotyping strategy and data, generated using tail tissue DNA samples, are shown. (**B**) Western blots, normalized to actin, and quantification for cortical and brainstem lysates from animals at P10 demonstrate cortex-restricted

*Figure 1. Continued*

suppression of GluN2B protein in 2BΔCtx mice (PFC = prefrontal cortex). Control lysates from P0 cortices of global GluN2B KO and WT animals run on the same blots served as positive and negative controls. Western blots and quantification of protein expression in control and 2BΔCtx cortex in vivo from P50–P70 animals shows selective decrease in GluN2B and no change in GluN1 and GluN2A. (**C**) Example images from NEX-Cre expressing DsRed flox-stop-flox GFP reporter mouse brain slices. Top, coronal section showing restricted cortical expression pattern of NEX-Cre (GFP+ = Cre+). Bottom, parasagittal slice showing strong Cre expression in both dorsal and ventral hippocampus and quantification, v = ventral, d = dorsal, scale bars = 500 µm. (**D**) Example traces recorded at +50 and −65 mV overlaid show the current response of layer II/III pyramidal neurons in control and 2BΔCtx slices from P18 to P21 animals in response to intracortical stimulation. Traces at +50 mV from control and 2BΔCtx slices demonstrate loss of ifenprodil sensitivity and faster decay kinetics of NMDAR-mediated current in 2BΔCtx neurons. (**E**) Combined analysis revealed a significant increase in the AMPAR to NMDAR-mediated current ratio in 2BΔCtx neurons, no change in overall peak current at +50 mV, a significant increase in the ifenprodil insensitive current at +50 mV, and a significant decrease in decay tau of the NMDAR-mediate current. Data values are means ± sem. *p < 0.05; **p < 0.01; ***p < 0.001 *t* test with respect to control; n.c. = no significant change.

results confirmed that GluN2B-containing NMDARs are absent in pyramidal PFC neurons in 2BΔCtx mice.

## Regulation of despair-like behavior by GluN2B in vivo

If GluN2B-mediated signaling, specifically in principal cortical neurons, is directly involved in regulating levels of despair-like behavior in response to NMDAR antagonism we would expect the 2BΔCtx genetic mutation to both mimic and occlude the actions of ketamine. To test this, we injected 2BΔCtx experimental and control animals with either ketamine or saline and then measured behavior and synaptic physiology in the same animals, 30 min and 25 to 30 hr later, respectively (*Figure 2A*). Consistent with our hypothesis, 2BΔCtx animals exhibited a dramatic decrease in despair-like behavior when compared to littermate control animals, as measured in two behavioral tests with strong predictive ability for antidepressant effectiveness; the forced swim test (FST) and tail suspension test (TST) (*Castagné et al., 2011*). In fact, the decrease in immobility scores in the FST test was so strong it precluded testing of whether or not the actions of ketamine injection were occluded in 2BΔCtx animals. Suppression of immobility scores in the TST by ketamine was occluded, as predicted, in 2BΔCtx mice and could be recapitulated by injection of the GluN2B-selective antagonist Ro 25–6981 (Ro) (*Figure 2B*). We next tested the sensitivity of these animals to chronic exposure to corticosterone (25 µg/ml). Surprisingly, while control animals exhibited a predicted increase in immobility in the TST following chronic corticosterone treatment (20 days of exposure), we saw no change in immobility scores in corticosterone-treated littermate 2BΔCtx animals (*Figure 2B*). These data show that 2BΔCtx animals are less susceptible to stress-associated changes in depression-like behavior.

In control animals changes in immobility measured in the TST were observed as early as 30 min after ketamine injection, consistent with previous reports (*Li et al., 2010*; *Autry et al., 2011*). 2BΔCtx animals also exhibited a strong and significant change in behavior in the elevated plus maze (EPM), which is less sensitive to changes in locomotor activity. Relative to control animals, which prefer the apparent safety of the closed arms of the maze, 2BΔCtx animals actually showed a reversal in this behavior, spending a significantly higher percentage of their time in the open arms of the maze, consistent with a strong reduction in anxiety in these mice (*Figure 2C*). Furthermore, subjecting animals to a chronic variable stress (CVS) paradigm resulted in a significant decrease in open arm time in control animals (Cont = 32.4 ± 10% vs Cont + CVS = 7.2 ± 1.7% p < 0.05) but no change in 2BΔCtx mice (2BΔCtx = 86.6 ± 6.7% vs 2BΔCtx + CVS = 69.3 ± 14.5% n.s.). Examining these mice in the open field test (OFT), we noted that changes in despair-like behavior in 2BΔCtx animals could not be accounted for by increased locomotor activity. Specifically, although total baseline locomotor activity is strongly increased in these 2BΔCtx animals, ketamine injection at this age caused a significant *suppression* of locomotion in both control and 2BΔCtx animals, indicating a dissociation of acute NMDAR suppression from locomotor activity, and consistent with previous data (*Figure 2D*) (*Akillioglu et al., 2012*). At this same time point immobility scores actually decreased (*Figure 2B*). Additionally, in food-restricted animals matched in age and mass to 2BΔCtx animals, we observed no significant change in immobility scores showing that the smaller body mass of these animals could not account for their decreased immobility times (*Figure 2E*). We examined reward-based behavior in the sucrose preference test in these animals,

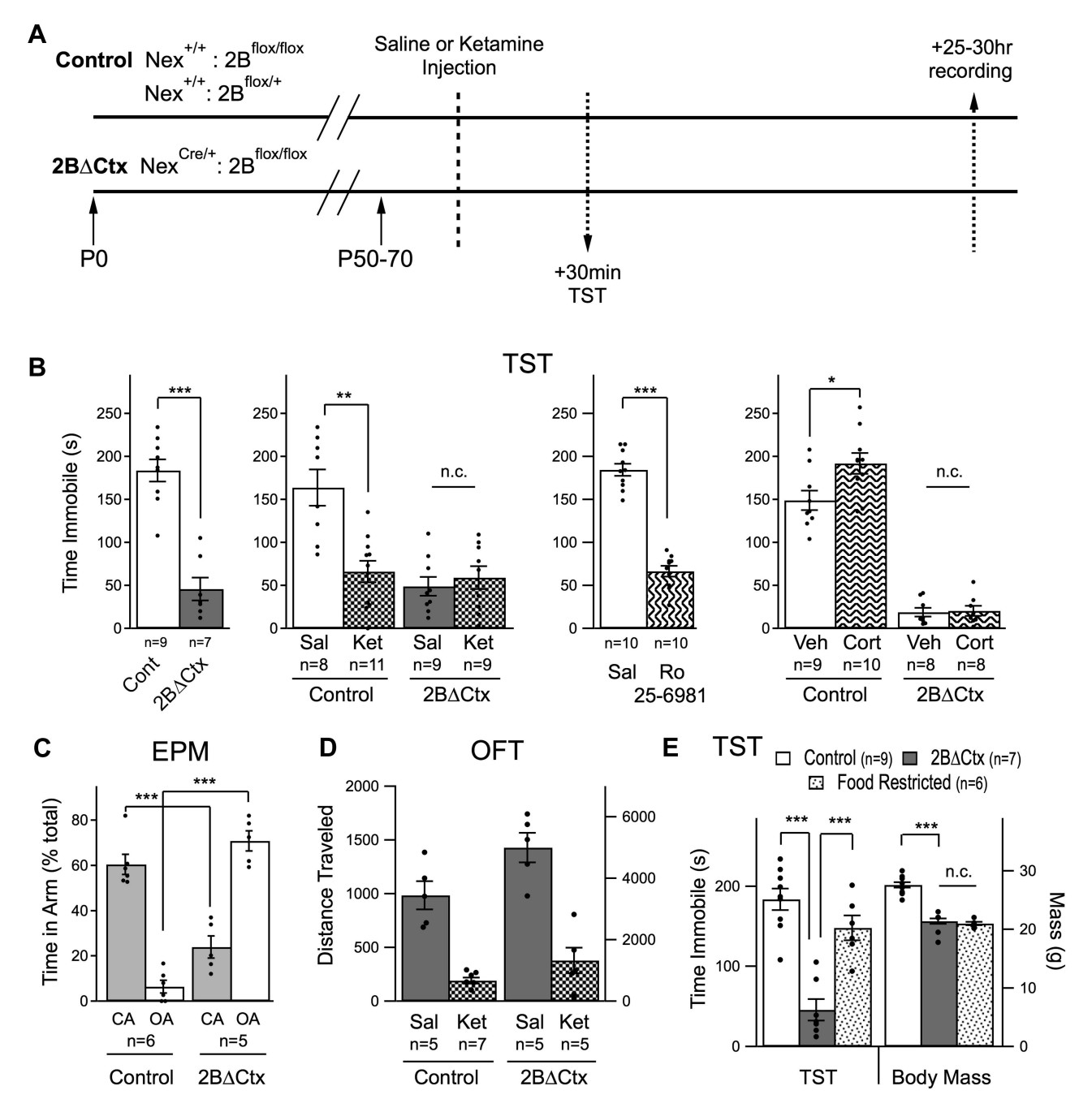

**Figure 2**. Decreased despair-like behavior and occlusion of ketamine's actions in 2BΔCtx animals. (**A**) Experimental timeline: male animals between P50 and P70 were subjected to i.p. ketamine injection (ket) or saline control injection (sal). 30 min following injection, animals were analyzed in the tail suspension test (TST). At 25–30 hr post-injection, animals were subjected to electrophysiological analysis (see **Figure 3**). (**B**) A significant decrease in immobility scores was measured in 2BΔCtx animals in the TST and this was mimicked by treatment with the GluN2B-containing NMDAR selective antagonist Ro 25–6981. The decrease in immobility scores in the 2BΔCtx animals occluded effect of ketamine injection seen in littermate control animals. 2BΔCtx animals were also insensitive to chronic corticosterone treatment (Cort), while this same treatment increased immobility times in TST in control, corticosterone-treated animals. (**C**) 2BΔCtx exhibited a strong anxiolytic behavioral phenotype compared to control animals as measured in the elevated plus maze (EPM). (**D**) Measuring total distance traveled in the open field test (OFT) showed a strong and significant effect of ketamine in both genotypes, suggesting that the decreased immobility time in 2BΔCtx animals was not simply due to hyperlocomotion. (**E**) Food restriction was also used in control animals to demonstrate that the decreased immobility times in 2BΔCtx animals was not a consequence of decreased body mass. Data values are means ± sem. *$p < 0.05$; **$p < 0.01$; ***$p < 0.001$ $t$ test with respect to control; n.c. = no significant change.

interestingly we measured no significant difference in the ratio of 10% sucrose water intake to regular water by 2BΔCtx mice compared to controls (Cont = 80.8 ± 4.7% vs 2BΔCtx = 70.5 ± 7.9% n.s.). The inability to detect a difference may reflect the high basal level of sucrose consumption in control animals or could reflect a dissociation of the domains of the behavioral phenotype in 2BΔCtx animals. It should be noted that cre is expressed strongly in the olfactory bulb of these conditional knockout animals, which could affect feeding behavior. Consistent with this idea, and their decreased body mass, food-deprived 2BΔCtx mice showed significantly higher latency to eat in a hidden food pellet task compared to littermate food-deprived control animals (Cont = 96.1 ± 19.4 s vs 2BΔCtx = 309 ± 74.4 s, p < 0.01). Our observation that 2BΔCtx mice had decreased motivation to feed based upon the hidden food pellet task as well as exhibiting longer latency to feed in non-anxiogenic, home-cage conditions (data not shown), in addition to their potentially disrupted olfactory sensitivity, complicated our ability to interpret data from a novelty suppressed feeding task (NSFT). In the NSFT, in a novel cage testing environment, we observed a strong trend to decreased latency to feed in control animals in response to ketamine injection (24 hr post-injection) as expected (Cont = 207.4 ± 70.6 s vs Cont + Ket = 58.7 ± 29.4 s, p = 0.17). Yet under basal conditions 2BΔCtx animals actually exhibited increased latency to feed compared to littermate control animals (Cont = 40.6 ± 7.1 s vs 149.1 ± 45.3 s, p = 0.06). Thus, we did not further test the effect of ketamine on these animals. These data generated from FST, TST, and EPM, both under basal conditions and in response to corticosterone treatment or chronic variable stress, strongly support the idea that GluN2B-mediated signaling, in cortical pyramidal neurons, is directly involved in setting basal levels of despair-like behavior in mice. Furthermore, loss of effectiveness of ketamine in 2BΔCtx animals indicates that GluN2B-mediated signaling might be required for ketamine's antidepressant actions.

## Increased excitatory synaptic transmission induced by ketamine in PFC is occluded in 2BΔCtx mice

We wondered whether or not increased synaptic activity, seen in response to ketamine injection, might be due to an increase in the number of individual excitatory synapses or modulation of the strength of existing synapses. We hypothesized that any changes observed in excitatory transmission in response to ketamine injection should be mimicked and occluded in 2BΔCtx animals, if they are in fact causally related to the behavioral phenotype. We focused on the PFC due to previous observations showing antidepressant-like actions of ketamine are antagonized by local infusion of rapamycin in PFC (*Li et al., 2010*), as well as the recognized importance of this structure in major depressive disorder (MDD) (*Price and Drevets, 2012*), and in response to behavioral challenge (*Warden et al., 2012*). Cortical excitatory synaptic transmission was examined by preparing acute brain slices from PFC 24 hr after vehicle or ketamine injection and applying whole-cell voltage clamp recordings at a holding potential of −65 mV while perfusing with TTX and picrotoxin to isolate miniature AMPAR-mediated excitatory synaptic currents (mEPSCs). These spontaneous synaptic currents result from individual neurotransmitter vesicle fusion events that are action potential independent, and therefore, allow assessment of increases or decreases in synaptic weight (amplitude) or changes in synapse number or probability of presynaptic transmitter release (frequency) under different experimental conditions. Our measurements showed that a single dose of ketamine increased excitatory synaptic neurotransmission onto layer II/III pyramidal neurons of control animals measured 24 hr after injection and this effect was mimicked and occluded in 2BΔCtx animals. Increased excitatory neurotransmission was measured as an increase in the frequency (decrease in inter-event interval) of mEPSCs in ketamine-injected animals relative to saline-injected controls (*Figure 3A–C*). Importantly, this ketamine-driven increase in mEPSC frequency was occluded in 2BΔCtx animals, which exhibited an increase in average baseline frequency of mEPSC events (*Figure 3A–C*). Behaviorally naïve animals also exhibited strong increase in mEPSC events 24 hr after ketamine injection, showing that this was not an effect of the behavioral testing (*Figure 3C*). In further support of a role for GluN2B signaling in ketamine's actions the selective antagonist Ro drove a similar increase in mEPSC events that correlated with decreased immobility in TST (*Figure 3C* c.f. 2B). The tight correlation between mEPSC frequency and immobility times in the TST strongly supports our hypothesis that excitatory synapse number in PFC contributes to setting basal levels of despair-like behavior. This prompted us to test whether or not this correlation could also be observed in corticosterone-exposed animals. Consistent with our hypothesis, we recorded a strong decrease in frequency of mEPSCs onto layer II/III pyramidal neurons in corticosterone-treated animals, compared to littermate vehicle exposed animals and this effect was absent in 2BΔCtx animals, where

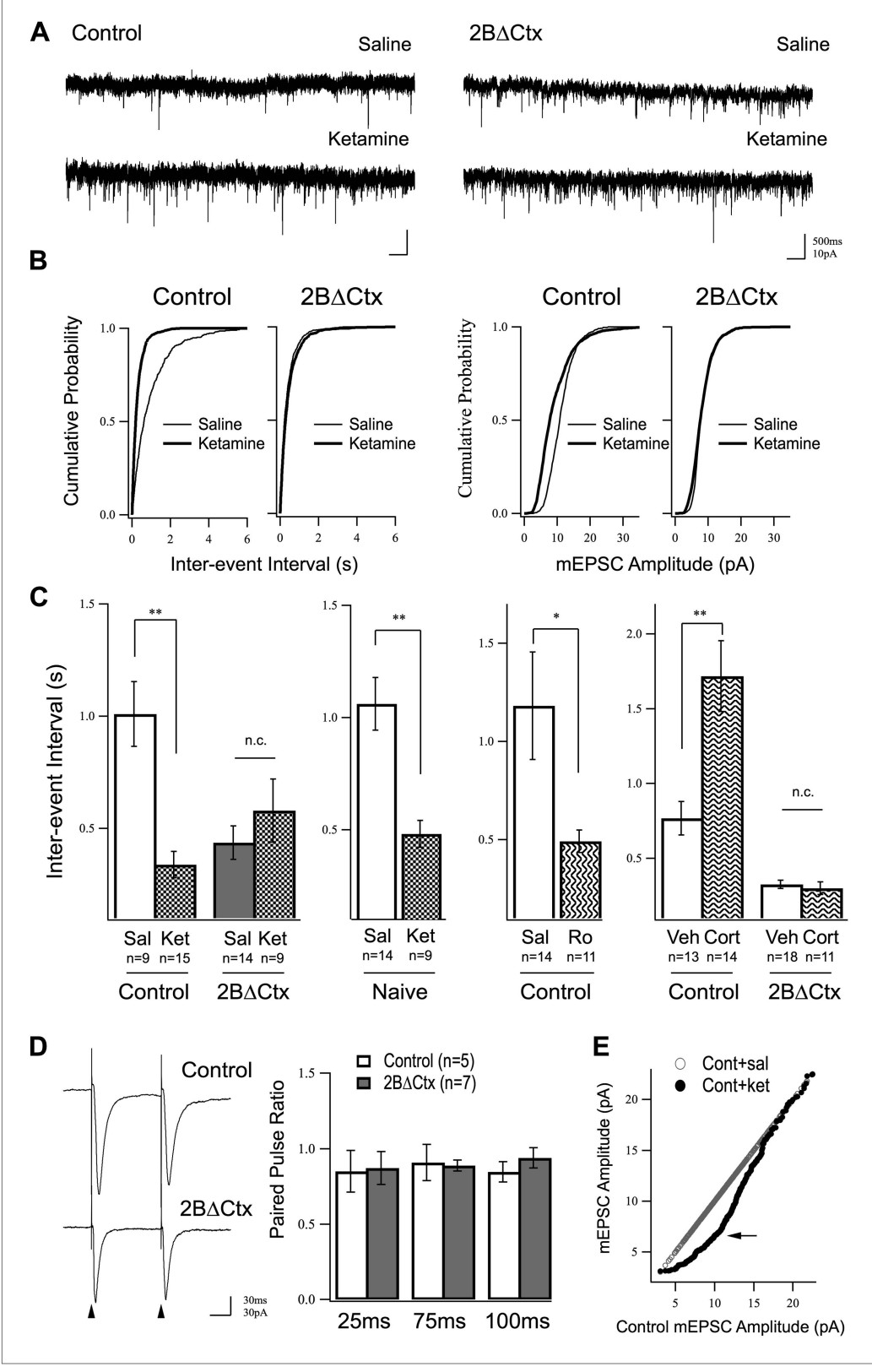

**Figure 3**. Increased excitatory synaptic transmission in prefrontal cortex following ketamine injection is occluded in 2BΔCtx animals. (**A**) mEPSC recordings from prefrontal layer II/III cortical pyramidal neurons 24 hr after a single injection of saline (upper) or ketamine (lower) in control and 2BΔCtx mice. (**B**) Cumulative histograms showing the

*Figure 3. Continued on next page*

*Figure 3. Continued*

strong increase in frequency (decreased inter-event interval) (left) of mEPSC events in control animals by ketamine is occluded in 2BΔCtx littermates. (**C**) Quantification and statistical IEI results are shown for data in (**B**). The increase in mEPSC frequency was not due to subjecting the animals to the behavioral testing as ketamine significantly suppressed IEIs in behaviorally naïve control animals. Additionally, 2BΔCtx animals showed no change in frequency of events measured after chronic corticosterone treatment, while this caused a strong decrease in event frequency in control animals. (**D**) The increase in event number in 2BΔCtx prefrontal cortical neurons was not correlated with a change in the paired pulse ratio measured by evoking synaptic responses at this synapse (intracolumnar stimulation indicated by the arrowheads). Example traces of evoked responses recorded by whole-cell voltage clamp at 100 ms inter stimulus interval are shown. Quantification across a number of inter-stimulus intervals is presented. (**E**) Ranked mEPSC plot showing the disproportionate increase in small amplitude events (arrow) in control ketamine-injected animals compared to saline injected controls. Data values are means ± sem. *p < 0.05; **p < 0.01; *t* test with respect to control; n.c. = no significant change.

baseline frequency of synaptic events was similar to ketamine-injected control animals (*Figure 3C*). Interestingly, on average, amplitude of mEPSC events in response to corticosterone treatment in either genotype were unchanged (Cont = 10.74 ± 0.66 pA; Cont + Cort = 9.58 ± 0.67 pA n.s.; 2BΔCtx = 10.50 ± 0.57 pA; 2BΔCtx + Cort = 10.46 ± 0.52 pA n.s.).

Increased mEPSC frequency could be due to an increase in synapse number or increase in probability of presynaptic neurotransmitter release. Surprisingly, the dramatic increase in mEPSC event frequency in 2BΔCtx animals was independent of a change in paired pulse ratio at intra-columnar II/III synapses (*Figure 3D*). While we did not test PPR at all synapses, and therefore, cannot rule out changes in presynaptic release at other contacts, our data are consistent with an increase in functional synapse number in line with previous recordings from GluN2B null neurons in hippocampal slices (*Gray et al., 2011*). The increase in synaptic event frequency in vivo was accompanied by a decrease in average mEPSC amplitude (Cont = 11.42 ± 0.97 pA vs Cont + Ket = 7.79 ± 0.98 pA p < 0.05). We infer this could either be due to a homeostatic down-regulation of synaptic strength in response to an increase in synapse number, or could be a reflection of insertion of young, immature synapses, which have low AMPAR content (*Harris and Stevens, 1989*; *Zito et al., 2009*). In support of this latter explanation, we saw a disproportionate increase in small amplitude events when comparing saline and ketamine injected event amplitude distributions in control animals (*Figure 3E*). Together, these data strongly support our hypothesis that ketamine suppresses GluN2B function in pyramidal cortical neurons leading to an increase in functional excitatory synaptic number in the PFC and a decrease in despair-like behavior.

## Suppression of protein synthesis by GluN2B

How are changes in synaptic physiology and behavior related to changes in protein synthesis and how does ketamine injection activate translational machinery in these cortical neurons in relation to GluN2B function? Previous results demonstrated a strong increase in cortical protein levels in response to ketamine however protein synthesis rates were not directly measured and the subunit specificity of NMDAR antagonism was not examined (*Li et al., 2010*; *Autry et al., 2011*). Our hypothesis predicts that loss of tonic suppression of the translational machinery by GluN2B-containing NMDARs would increase protein synthesis rates in GluN2B null neurons. To determine whether or not GluN2B-containing NMDARs act to directly suppress protein synthesis in pyramidal cortical neurons, we prepared cultures from WT and global GluN2B KO (*Kutsuwada et al., 1996*) cortices and analyzed rates of translation using fluorescence-based non-canonical amino acid tagging (FUNCAT) (*Dieterich et al., 2010*). Incorporation of the non-canonical amino acid AHA, which substitutes for methionine in nascent peptides during translation, was used as a measure of protein synthesis rates. After a controlled, 6-hr incubation in AHA-containing methionine-free media, neurons were fixed and AHA was fluorescently labeled. Anti-MAP2 immunostaining was used to delineate dendritic segments for analysis (*Figure 4A*). We observed a dramatic increase in the rate of protein synthesis in GluN2B KO neurons compared to controls, consistent with our hypothesis (*Figure 4B*). However, at these ages (11–14 DIV) in rodent cortical cultures both GluN2A and GluN2B protein is expressed (*Li et al., 1998*; *Hall et al., 2007*). We therefore wanted to know if the increase in protein synthesis in GluN2B KO cultures was due to loss of GluN2B *specifically*, or reduction in NMDAR-mediated current *generally*.

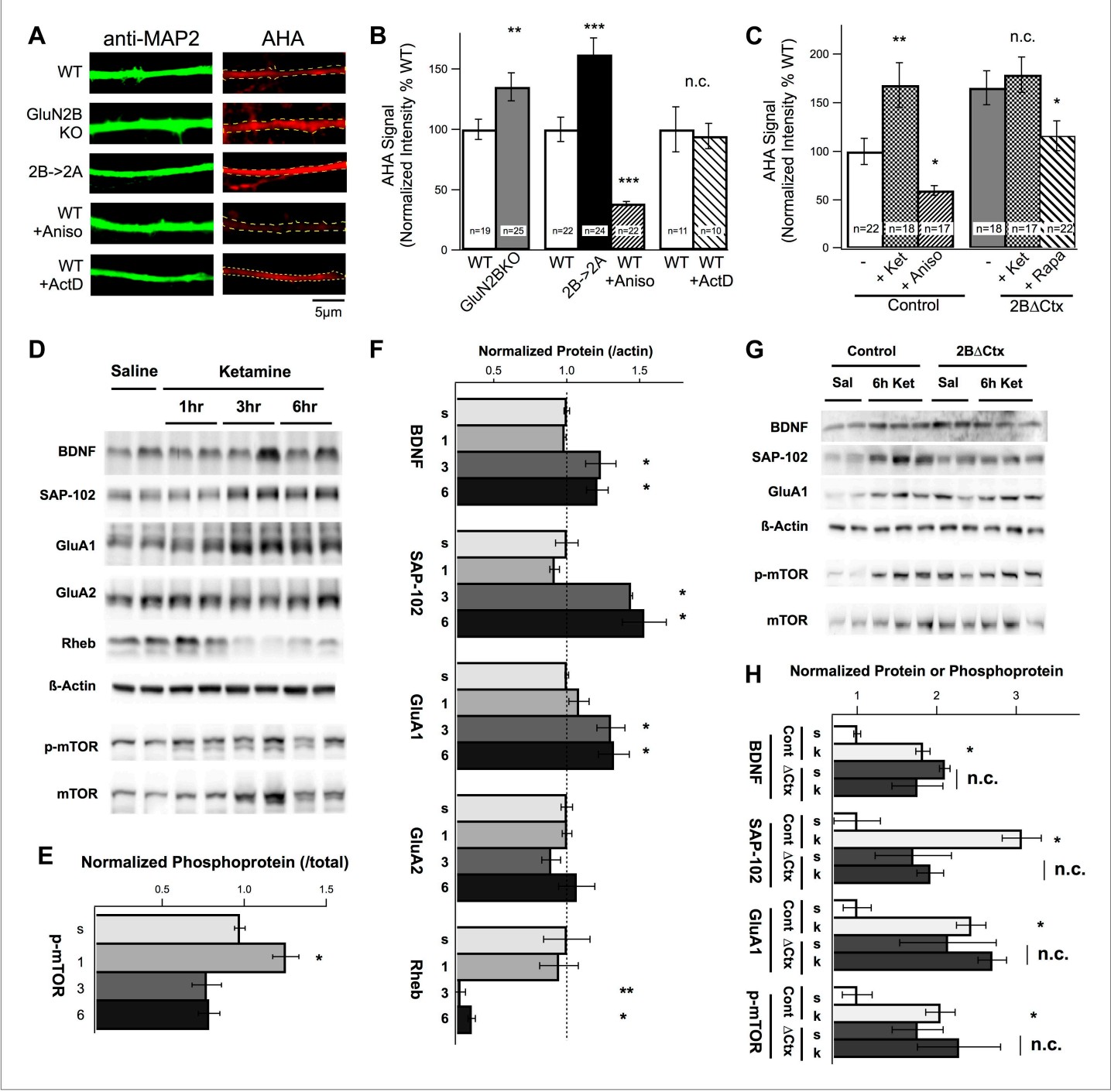

**Figure 4**. Changes in protein synthesis rates, synaptic protein expression, and phosphorylation in response to ketamine injection are occluded in GluN2B null neurons. (**A**) FUNCAT was used to measure rates of protein synthesis in cortical neurons. AHA signal intensity in MAP2-stained dendrites reveals relative levels of new protein synthesized over a 6-hr period. Examples of 14 DIV cortical neurons from GluN2BKO and 2B→2A as well as WT control neurons treated with the protein synthesis inhibitor anisomycin (Aniso) or transcription inhibitor actinomycin-D (ActD). (**B**) Combined data show significant increase in protein synthesis rates in GluN2BKO and 2B→2A neurons and suppression of basal levels by anisomycin. (**C**) Combined data showing the significant increase in protein synthesis rates evoked by ketamine, occlusion of this increase in 2BΔCtx and sensitivity of the AHA signal to rapamycin in 2BΔCtx neurons. (**D–H**) Cortical synaptoneurosomes from animals following saline or ketamine injection. Western blot analysis showed the presence of mTOR in synaptoneurosomes as well as basal levels of phosphorylated protein (p-mTOR) and expression of BDNF, Rheb, and the synaptic proteins, SAP-102, GluA1, and GluA2. (**D**) Expression levels, and phosphorylation status of mTOR, following ketamine injection (1, 3, and 6 hr post injection) compared to saline-injected controls. (**E**) p-mTOR measured relative to total mTOR levels demonstrating time-dependent changes in phosphorylation

*Figure 4. Continued on next page*

Figure 4. Continued

status in response to ketamine. (**F**) Synaptic protein expression is shown in relation to levels of actin at 1, 3, or 6 hr post ketamine normalized to saline-injected controls. Increased levels of GluA1, SAP102, and BDNF as well as decreased rheb expression seen at 3 and 6 hr post injection. (**G**) Example of western blots at 6 hr post-injection for both control and 2BΔCtx genotypes. (**H**) Quantification of the data showed significant increases in synaptic proteins in response to ketamine (k), consistent with (**A–C**), but also revealing occlusion of these increases in SAP-102, GluA1 and p-mTOR in 2BΔCtx animals. Data values are means ± sem. **$p < 0.01$; ***$p < 0.001$ $t$ test with respect to WT; n.c. = no significant change.

To test this, we generated neuronal cultures from cortices of animals in which GluN2B had been genetically replaced by GluN2A (2B→2A) (*Wang et al., 2011a*). In homozygous 2B→2A cultures, genetic replacement of GluN2B by GluN2A was not sufficient to restore tonic suppression of translation, confirming that this is a unique function of GluN2B-containing NMDARs (*Figure 4B*). These data provide direct evidence that GluN2B acts in a unique manner, under conditions of basal activity, to tonically suppress protein synthesis in cortical neurons in vitro. However, if ketamine antagonizes this cellular action, we should be able to increase protein synthesis rates by applying ketamine to control cultures. Indeed, ketamine caused a significant increase in AHA signal in dendrites of control neurons compared with non-treated cells and this increase was mimicked in 2BΔCtx neurons in a manner that was sensitive to rapamycin treatment (*Figure 4C*). The inability of ketamine to further increase protein synthesis rates in 2BΔCtx neurons is highly suggestive that GluN2B-containing receptors are required for the effects of ketamine (*Figure 4C*). Next, we asked what cellular mechanisms in vivo were responsible for these changes in protein synthesis in response to ketamine, and whether or not these changes could be mimicked by GluN2B loss of function, in vivo, in 2BΔCtx animals.

## Ketamine-induced increase in protein synthesis, in vivo, is occluded in 2BΔCtx animals

To examine the mechanisms through which ketamine regulates the translational machinery in vivo and determine its regulation by GluN2B, we prepared synaptoneurosomal fractions from both control and 2BΔCtx animals at increasing times after injection and performed quantitative western blot analysis. Western blot analysis revealed mTOR expression in these biochemical synaptic fractions, predicting a close association between mTOR and synaptic sites (*Scheetz et al., 2000*; *Li et al., 2010*) (*Figure 4D*). In response to ketamine injection, we observed a rapid yet transient increase in phosphorylated mTOR (p-mTOR), relative to total mTOR (*Figure 4D,E*). Interestingly, the late-phase decrease in mTOR activation we observed coincided with decreased expression of the upstream mTOR activator Rheb, suggesting a possible negative-feedback mechanism (*Figure 4D,F*). Expression of a number of synaptic proteins in these synaptic fractions was increased, including brain-derived neurotrophic factor (BDNF), synapse associated protein 102 (SAP-102), and the AMPAR subunit GluA1 (*Figure 4F*). This is consistent with the increase in excitatory synapse number predicted by our mEPSC analysis. BDNF has been shown to regulate increases in synapse number and synapse unsilencing (*Itami et al., 2003*; *Shen and Cowan, 2010*) and a rapid increase in the number of immature synaptic connections is consistent with the increased expression of SAP-102 and GluA1 protein, which are associated with developmentally young synapses (*Pellegrini-Giampietro et al., 1992*; *Martin et al., 1998*; *Sans et al., 2001*; *van Zundert et al., 2004*). Because 2BΔCtx both mimics and occludes the behavioral actions of ketamine, alterations in protein expression that are causally associated with these effects should be mimicked and occluded by GluN2B loss of function. In 2BΔCtx synaptoneurosomes, we observed increased baseline levels of BDNF, SAP-102, GluA1, and p-mTOR and increased basal levels of expression of these proteins (and phosphorylation of mTOR) occluded any further increase in response to ketamine injection in 2BΔCtx animals (*Figure 4G,H*). In addition to p-mTOR, we also observed strong increases in p-P70S6K in both 2BΔCtx PFC lysates and 2A→2B lysates from P50–P70 animals compared to controls (Cont = 100 ± 13.3% vs 2BΔCtx = 217 ± 11.4% $p < 0.01$; WT = 100 ± 34% vs 2B→2A = 318 ± 55% $p < 0.05$). These data strongly suggest that these increases in protein expression are driven by mTOR activation and causally related to the actions of ketamine and that these levels are regulated by GluN2B-containing NMDARs.

## GluN2B-containing NMDARs are uniquely activated by ambient glutamate

The above data support a predicted requirement for GluN2B-mediated signaling in the actions of ketamine, yet an important question remains, how does a *non-selective* NMDAR antagonist like

ketamine produce a seemingly *selective* effect on GluN2B-mediated signaling? Despite the fact that GluN2B-selective antagonists have been shown to have antidepressant-like actions in both clinical and preclinical studies, the exact role of GluN2B-containing NMDARs has remained unclear. From our FUNCAT data, we infer that GluN2B-containing NMDARs function under non-stimulated conditions to tonically suppress protein synthesis in a manner that is not rescued by replacement with GluN2A. From these observations, we hypothesized that GluN2B-containing NMDARs might be selectively activated under basal, non-stimulated, conditions. If this receptor pool is uniquely tonically active, this would explain why either GluN2B-selective or non-selective pan-NMDAR antagonists have the same effect of promoting protein synthesis. Consistent with this idea, spontaneous, non-synchronized transmitter release has been shown to be sufficient in vitro to limit protein synthesis in cortical neurons including through suppression of mTOR (**Sutton et al., 2007**; **Wang et al., 2011a**). However, in addition to synaptically released glutamate, there is strong evidence supporting the presence of persistent, low-level ambient glutamate in vivo and in brain slices (**Meldrum, 2000**). Interestingly, changes that are consistent with increased ambient glutamate, including glial retraction and decreased expression of glutamate transporters, have been demonstrated in human postmortem studies of depressed patients and observed in animal models of depression (**Moghaddam et al., 1994**; **Boudaba et al., 2003**; **Rajkowska and Miguel-Hidalgo, 2007**; **Banasr and Duman, 2008**; **Zink et al., 2010**).

We therefore wondered if ambient glutamate could activate a tonic NMDAR-mediated current in cortical neurons and whether or not this current is mediated selectively by GluN2B-containing NMDARs? To examine this, we first perfused WT-cultured cortical neurons with TTX, to block synchronized transmitter release, and DNQX and picrotoxin to block AMPAR and GABA$_A$R currents, respectively, then washed in 0Mg$^{2+}$ ACSF. This revealed a tonic current evident as a reliable increase in the signal noise of the holding current (root mean square signal—RMS) at V$_{hold}$ −65 mV. The increase in tonic current was suppressed by APV, demonstrating its dependence upon NMDAR activation, and by ifenprodil, supporting the prediction that GluN2B-containing NMDARs are selectively activated by ambient glutamate (**Figure 5A**). To further verify the subunit specificity of this NMDAR-mediated response, we next examined its presence in our genetically modified neurons. Strikingly, in a manner consistent with selective activation of GluN2B-containing NMDARs, the ambient glutamate-activated tonic current was absent in GluN2B null cultures, as well as in cells where NMDARs levels have been restored but GluN2B was genetically replaced by GluN2A (**Figure 5A**). This tonic current could also be evoked in acute cortical slices and was completely antagonized by ketamine application or by the GluN2B selective antagonist ifenprodil but was not suppressed by Zn$^{2+}$, which is a strong and selective GluN2A antagonist at low concentrations (250 nM) (**Figure 5B**). We also saw that the tonic current activated by ambient glutamate in 0Mg$^{2+}$ was lost in 2BΔCtx slices (**Figure 5B**). Finally, we generated mosaic GluN2B knockout conditions in PFC by in vivo injection of a GFP-Cre expressing AAV virus under the CaMKII promoter into homozygous conditional GluN2B KO animals and recorded tonic current in acute brain slices. In these GluN2B null neurons, we also noted that 0Mg$^{2+}$ conditions generated no increase in RMS signal yet GluN2A-containing receptors could be activated under these conditions as perfusion of these slices with 25 μM NMDA in 0Mg$^{2+}$ saline, resulted in a strong increase in RMS signal (baseline ACSF = 1.29 ± 0.18, n = 7; 0Mg$^{2+}$ = 1.25 ± 0.15, n = 11; 0Mg$^{2+}$ + 25 μM NMDA = 2.88 ± 0.60, n = 5; p < 0.001 c.f. baseline). Furthermore, the average holding current increased to 352 ± 103.9 pA (n = 3) after 100 s of continued slow perfusion with NMDA. These data confirm that ambient glutamate is sufficient (both in acute brain slices and primary neuronal cultures) to activate a uniquely sensitive population of GluN2B-containing NMDARs on cortical neurons that can be antagonized by ketamine.

Next, we used excitatory amino acid transporter (EAAT) inhibitors and enhancers to manipulate the tonic current and test its ability to regulate mEPSC frequency. As shown in **Figure 5C**, pre-treatment with the glutamate transporter antagonist dl-TBOA resulted in a significant increase in the tonic current evoked in 0Mg$^{2+}$, while the EAAT enhancer nordihydroguaiaretic acid (NDGA) (**Boston-Howes et al., 2008**) significantly suppressed the strength of this GluN2B-mediated tonic current. Because our in vivo data predict an inverse relationship between GluN2B activation and synapse number, we then determined whether or not manipulation of this current in cultured neurons alters mEPSC frequency. We pretreated cultures with dl-TBOA, to increase ambient glutamate concentration, or with NDGA or ceftriaxone (24 hr and 1 week, respectively), to enhance EAAT function and decrease ambient glutamate (**Rothstein et al., 2005**; **Rasmussen et al., 2011**). Unfortunately, chronic NDGA application turned out to be lethal to the cultured cells so only ceftriaxone was testable in the chronic experiments.

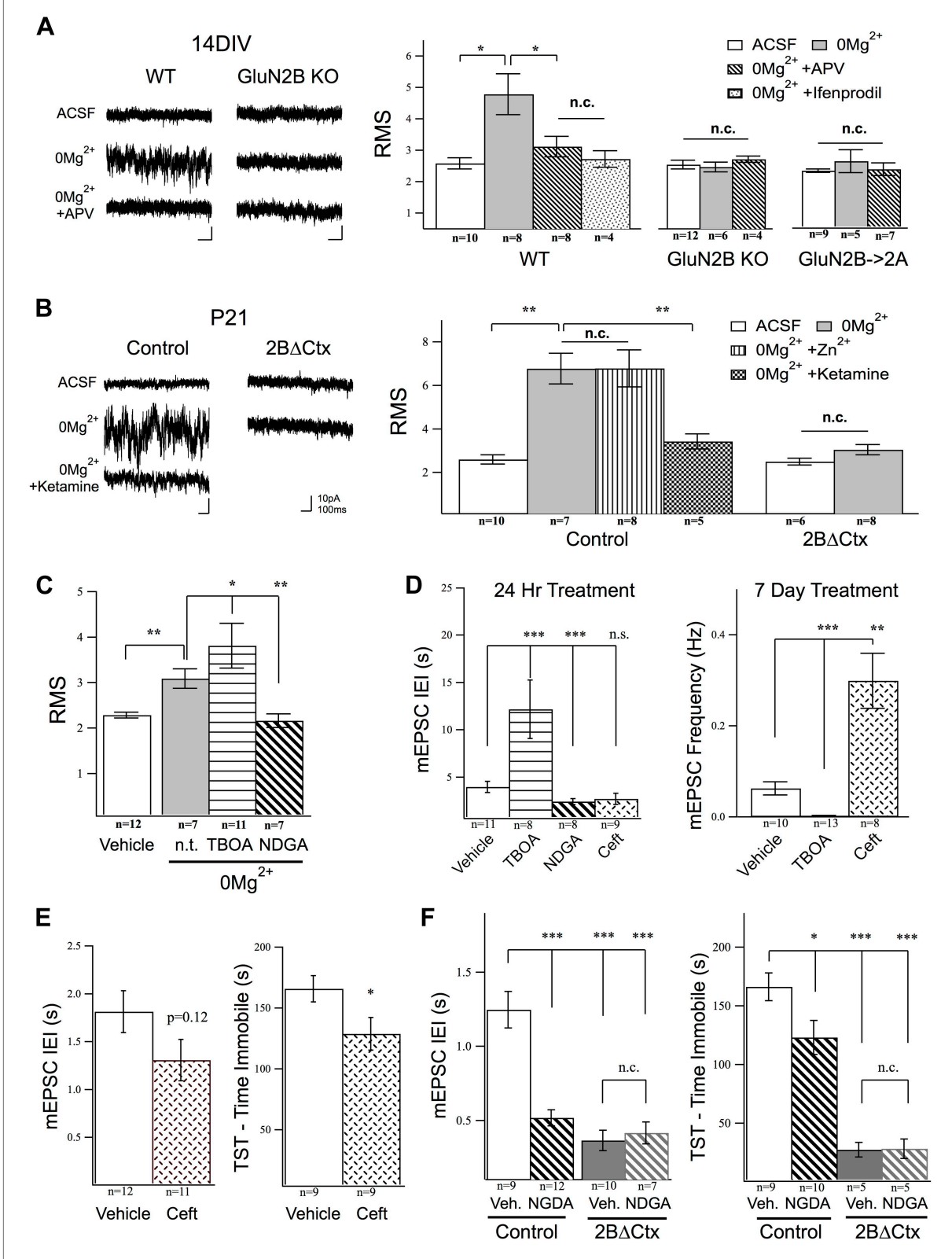

**Figure 5**. Activation of GluN2B-containing NMDARs by ambient glutamate regulates mEPSC frequency and depression-like behavior. Whole-cell recordings of cortical pyramidal neurons ($V_{hold}$ −65 mV) under control conditions (ACSF) and in $0Mg^{2+}$ ACSF were used to test the subunit contribution of the NMDA receptor pool activated by ambient glutamate. (**A**) Perfusion of $0Mg^{2+}$ ACSF uncovered a tonic current in cultured cortical neurons (14 DIV)

*Figure 5. Continued on next page*

*Figure 5. Continued*
evidenced by an increase in baseline current noise that could be reversed by the NMDAR antagonist APV. This tonic NMDAR-mediated current was completely absent in GluN2B null neurons, including those in which GluN2B had been genetically replaced by GluN2A (GluN2B→2A). (**B**) Ambient glutamate also evoked a tonic current in acute brains slices that was suppressed by ketamine and was absent in 2BΔCtx slices (P21). (**C–D**) Manipulation of tonic activation of GluN2B-containing NMDARs alters mEPSC frequency in cortical cultures and pyramidal neurons in vivo, as well as expression of depression-like behavior. (**C**) Acute treatment of cultures with the glutamate transporter antagonist dl-TBOA enhanced the evoked current, while NDGA, which enhances glutamate transporter function, decreased this current. (**D**) Elevating ambient glutamate by blocking transporters with dl-TBOA, or decreasing it by enhancing glutamate transporter function with NDGA, or upregulating glutamate transporter expression using ceftriaxone bidirectionally regulated mEPSC frequency in cultured cortical neurons. (**E**) Enhancing glutamate transporter expression by i.p. injection of ceftriaxone resulted in a decrease in mEPSC IEI, and significant antidepressant-like action in control mice. (**F**) Injections of NDGA resulted in a significant decrease in mEPSC IEI in pyramidal neurons and decreased immobility in the TST. The effect of NDGA was mimicked and occluded on both measures in 2BΔCtx mice. Data points are mean ± sem. p-values are *<0.05, **<0.01, ***<0.001, and n values are shown for each experiment. n.c. = no significant change.

In line with our prediction, the frequency of mEPSC events measured in pyramidal cortical neurons increased in response to 24 hr NDGA treatment and decreased dramatically in response to antagonism of glutamate re-uptake by dl-TBOA (*Figure 5D*). At this timepoint, however, ceftriaxone caused no significant increase in mEPSC frequency, perhaps owing to the mechanism of its action, which involves increasing translation and membrane insertion of EAATs, rather than direct enhancement of existing transporters. However, after 1 week of treatment ceftriaxone-treated cells showed dramatically increased mEPSC frequency. Cells treated with TBOA, which were otherwise healthy, were silenced after 7 days of treatment (*Figure 5D*). These data show that regulation of tonic glutamate levels can affect synapse function in cortical networks and that modulation of extracellular glutamate uniquely activates GluN2B-containing NMDARs to regulate excitatory synaptic drive in cortical neurons. Based upon our in vitro results, we therefore hypothesized that enhancing glutamate transporter function should suppress depression-like behavior and increase mEPSC frequency, in vivo. To test this, we injected animals with either NDGA (acute) or Ceftriaxone (chronic) to increase EAAT function or expression, respectively (*Rothstein et al., 2005*; *Mineur et al., 2007*; *Boston-Howes et al., 2008*). Supporting our model, we observed decreased depression-like behavior in TST in these experimental animals compared to vehicle injected control animals, as well as a corresponding increase in mEPSC frequency in PFC (*Figure 5E,F*). Taken together these data strongly supporting a role for tonic activation of GluN2B-containing NMDARs in setting basal levels of despair-like behavior.

## Discussion

Based upon these data, we propose that tonic activation of GluN2B-containing NMDARs suppresses protein synthesis in cortical pyramidal neurons and loss of this results in increased protein synthesis, an increase in the number of excitatory inputs onto pyramidal neurons in PFC, and a decrease in despair evidenced, in mice, by increased motivation in TST and decreased anxious behavior in EPM. Our data suggest this involves regulation of mTOR downstream of GluN2B. Consistent with our hypothesis that these receptors are involved in the rapid actions of ketamine, restricted genetic removal of GluN2B from cortical pyramidal neurons decreased basal despair-like behavior, occluded the actions of ketamine, and suppressed the increase in depression-like behavior associated with chronic exposure to corticosterone. These studies extend our understanding of the mechanisms underlying the rapid anti-depressant actions of ketamine while providing a novel genetic animal model for depression studies. Importantly, they provide a novel mechanistic framework for interpreting the relationships between ambient glutamate, basal GluN2B-signaling, excitatory cortical neurotransmission, and behavioral despair.

Conventional antidepressant treatments are effective in approximately one-third of major depressed patients and in the responding population standard selective serotonin reuptake inhibitor (SSRI) therapy exhibits a pronounced delay to efficacy (*Trivedi et al., 2006*). This observation led to the 'initiation and adaptation' hypothesis of depression treatment (*Hyman and Nestler, 1996*). According to this theory, treatment initiates an adaptive response leading to the delayed, yet consequential, effects of pharmacological intervention. This may involve alterations in NMDAR function. Consistent with a role for NMDARs in MDD, inescapable stress has been shown to lead to disruptions in NMDAR-dependent synaptic plasticity concurrent with induction of behavioral depression-like behavior (*Trullas and Skolnick, 1990*). Moreover, SSRI treatment can lead to an adaptation of the NMDAR-mediated

response (**Skolnick et al., 1996**), decreases in expression of NMDAR subunits (**Nowak et al., 1996**) and expression of NMDAR encoding mRNAs (**Boyer et al., 1998**).

The mechanism through which NMDAR antagonism exerts rapid antidepressant actions has therefore become an important direction of study (**Duman et al., 2012**). A critical question is how does *suppression* of NMDAR function *promote* protein synthesis to enhance synaptic function? Previous explanations have proposed circuit-level hypotheses in which NMDAR antagonism preferentially acts upon inhibitory GABAergic neurons leading to an *indirect* increase in firing in principal neurons, release of BDNF, and increased synaptic strength (**Stefani and Moghaddam, 2005**; **Duman et al., 2012**). Our data suggest that ketamine acts *directly* on principal cortical neurons to remove basal suppression of mTOR-mediated protein synthesis by antagonism of GluN2B-containing NMDARs. In support of this, our data reveal strong suppression of depression-like behavior in mice lacking GluN2B specifically in pyramidal cortical neurons. These two mechanistic explanations are not mutually exclusive and could actually work together: initial increases in protein synthesis due to suppression of NMDAR function activates mTOR and promotes an increase in synapse number in PFC while increased network activity, increased glutamate release, or repeated activation of these contacts by spontaneous activity, might be required for stabilization (**Moghaddam et al., 1997**; **McKinney et al., 1999**). Consistent with this model, it is known that antagonism of AMPARs, which would prevent stabilization of new contacts, blocks the antidepressant actions of ketamine (**Maeng et al., 2008**; **Koike et al., 2011**). We also have preliminary data to support this idea as we find that the AMPAR antagonist DNQX does not block the rapid increase in FUNCAT signal (i.e., increased protein synthesis) after ketamine exposure but does block the increase in spine density (i.e., stability of contacts) in, in vitro cortical cultures, 24 hr after exposure to ketamine. Also in line with this mTOR based model is the understanding that GSK-3 inhibition promotes ketamine's actions via augmentation of mTORC1 activation, while constitutively active GSK-3 mice are insensitive to ketamine's antidepressant-like actions (**Li et al., 2002**; **Beurel et al., 2011**; **Liu et al., 2013**). It will be interesting in future experiments, therefore, to test the relative roles of these pathways in regulating excitatory synapse number in PFC and behavior while exploring potential links between GluN2B and GSK-3-mediated signaling. In terms of additional downstream effectors, we have also shown recently that protein synthesis can be suppressed by SynGAP signaling via inhibition of mTOR and that this is unique to GluN2B-containing NMDARs (**Wang et al., 2013**).

Examining the time-course of mTOR activation in synaptic fractions of cortical lysates following in vivo ketamine injection, we observed that the initial increase in mTOR activation is transient. Interestingly, we also observed a decrease in the mTOR upstream activator Rheb in ketamine-treated cortex 3–6 hr after ketamine injection. Rheb is an immediate early gene and a strong activator of the mTOR kinase (**Yamagata et al., 1994**; **Garami et al., 2003**; **Laplante and Sabatini, 2012**). This suggests that Rheb itself may be regulated at a translational level, which is supported by the observation that Rheb mRNA is present in forebrain neuronal dendrites (**Cajigas et al., 2012**). Additional studies will be required to determine the full complement of proteins whose expression is altered in response to ketamine. Our data show increased levels of SAP-102 and GluR1 following ketamine injection, suggesting a rapid increase in new, excitatory synapses in layer II/III of prefrontal cortex since these proteins are associated with young synapses (**Pellegrini-Giampietro et al., 1992**; **Martin et al., 1998**; **Sans et al., 2001**; **van Zundert et al., 2004**). BDNF signaling and mTOR activation enhance synaptogenesis and promote synapse unsilencing in cortex (**Itami et al., 2003**; **Luikart and Parada, 2006**; **Hoeffer and Klann, 2010**; **Shen and Cowan, 2010**). In addition, BDNF can enhance pre-synaptic function (**Henry et al., 2012**). Our biochemical data show mTOR is localized close to synapses, as evident by its presence in our synaptoneurosome preparations. This is in line with previous reports (**Scheetz et al., 2000**; **Li et al., 2010**) and implies that protein synthesis, in response to ketamine, might be occurring locally in dendritic compartments. Local protein synthesis maintains proper levels of synaptic strength in cortical and hippocampal neurons and this regulation has been implicated in regimes of homeostatic synaptic plasticity (**Ju et al., 2004**; **Sutton et al., 2006**; **Aoto et al., 2008**; **Wang et al., 2011a**). In light of this, other cellular regulators of dendritic protein synthesis and homeostatic synaptic plasticity need to be examined in relation to ketamine's actions, including mRNA binding proteins FMRP (**Ashley et al., 1993**), CPEB (**Richter, 2007**), and translational regulators retinoic acid (**Wang et al., 2011b**), and eIF4E (**Gkogkas et al., 2012**).

Interestingly, GluN2B knockout induced by CaMKII-promoter-driven Cre expression results in a weaker phenotype than the NEX-Cre phenotype reported here (**Von Engelhardt et al., 2008**;

*Kiselycznyk et al., 2011*). This phenotypic discrepancy, if not due to sex, age, or mouse strain differences, is most likely due to differences in the spatial expression pattern of Cre-recombinase protein. While NEX-Cre results in developmental excision of GluN2B and CaMKII-promoter driven Cre does not (*Tsien et al., 1996*; *Goebbels et al., 2006*), this is unlikely to be solely responsible for the different phenotypes since the antidepressant actions of ketamine are evoked by acute treatment (*Preskorn et al., 2008*). This suggests that the specificity of the genetic knockout in cortical structures is responsible for this discrepancy. Indeed, cortical lysates from CaMKII-driven Cre animals contain more residual GluN2B protein, supporting the interpretation of a wider distribution of Cre expression in 2BΔCtx animals (*Brigman et al., 2010*). Another notable difference is that Cre-driven gene excision is restricted to the dorsal hippocampus in the CaMKII animal (*Tsien et al., 1996*). Suppression of GluN2B function in the ventral hippocampus of Nex-Cre based 2BΔCtx animals (*Figure 1C*) likely contributes to the behavioral phenotype we observe as activity in ventral dentate strongly regulates anxiety (*Kheirbek et al., 2013*). Future experiments will require more precise targeting of specific brain regions and cell types to determine their role in aspects of the GluN2B KO phenotype. Of course, potential developmental changes induced by GluN2B knockout should not be ruled out especially in light of the importance of circuit maturation changes in vulnerability to mood disorders (*Stefani and Moghaddam, 2005*; *Ansorge et al., 2007*). The absence of a positive effect on sucrose intake in the SPT in the 2BΔCtx animals is also notable, as this hedonic behavior is considered a domain of the depression-like phenotype in preclinical models. Interestingly, recent elegant experiments have shown a critical and specific role for synaptic transmission in the nucleus accumbens in sucrose intake in SPT in mice (*Lim et al., 2012*). The absence of any Cre expression in this nucleus in the 2BΔCtx mice (data not shown) may explain this observation while also supporting the idea that these phenotypic domains can be separated in preclinical models of depression.

There is strong evidence supporting the presence of persistent, low-level ambient glutamate both in vivo and in brain slices (*Meldrum, 2000*) and NMDARs can be activated by ambient glutamate in hippocampal neurons (*Sah et al., 1989*; *Papouin et al., 2012*). Our data show that a tonic NMDAR-mediated current can be evoked in cortical neurons both in culture and acute brain slices. Strikingly, this current is absent in GluN2B null cortical neurons and in neurons in which GluN2B has been genetically replaced with GluN2A (2B→2A). This shows that endogenous GluN2B receptors are more sensitive to ambient glutamate than those containing GluN2A. Indeed, GluN2B-containing NMDARs have higher sensitivity to agonist, a shifted $Mg^{2+}$ sensitivity (*Mori and Mishina, 1995*) and higher concentrations of NMDA and d-serine are required to evoke equal current in neurons expressing only GluN2A-containing NMDARs (*Wang et al., 2011a*). Furthermore, it is still unclear exactly how endogenous proteins regulate NMDAR function in situ, compared to heterologous expression studies. Levels of ambient, extracellular glutamate are tightly regulated by glutamate transporters especially EAAT2 (GLT-1), which is highly expressed on glial cells (*Rothstein et al., 1996*; *Guo et al., 2003*). While the absolute concentration and role ambient glutamate is highly debated, our data show that basal activation of GluN2B signaling is functionally relevant and can tonically suppress protein synthesis in these neurons (*Figure 4*). Furthermore, in support of a role for tonic NMDAR signaling in regulating excitatory synapse function, we show that mEPSC frequency changes in a predicted manner when glutamate transporter function is chronically altered under standard $Mg^{2+}$ in cortical cultures maintained at 37°C (*Figure 5*). Thus, we propose that ambient glutamate suppresses protein synthesis in a GluN2B-dependent manner in order to maintain synapse number in cortical neurons and thereby contribute to setting levels of despair-like behavior. This model is consistent with the strong association between decreased glial/glutamate transporter function and depression. This includes evidence that EAAT expression is decreased by exposure to learned-helplessness in rats (*Zink et al., 2010*), a decrease in glial density observed in the PFC of chronically stressed animals (*Banasr and Duman, 2008*), decreased glial density in postmortem assessment of PFC in depressed subjects (*Rajkowska and Miguel-Hidalgo, 2007*), glial retraction from synapses induced by stress (*Boudaba et al., 2003*), and induction of depression-like behavior through chemical ablation of glial cells in PFC in mouse (*Banasr and Duman, 2008*). The strongest support for this hypothesis comes from experiments in which transporter function is manipulated in vivo. As our data show, acute increase in EAAT function (NDGA) and increasing EAAT expression (ceftriaxone) in vivo result in decreased immobility in TST and a corresponding increase in frequency of mEPSCs in layer II/III pyramidal neurons of the PFC (*Figure 5E,F*). NDGA and ceftriaxone have multiple functional effects; however, since they have no published common targets (other than in enhancing EAAT2-mediated glutamate uptake), the most likely explanation we have at

this time is that they are both acting through a common pathway to enhance tonic GluN2B activation and cause the resultant downstream effects we observe in behavior.

Future studies need to focus on determining the location of the GluN2B-containing receptor pool activated by ambient glutamate and the in vivo conditions under which they are tonically activated, especially under pathophysiological conditions. Since this receptor pool is uniquely tied to regulation of protein synthesis, it could prove to be an important target for antidepressant therapy and other disorders associated with dysregulation of protein synthesis. Recent reports have shown that NMDARs are activated by distinct co-agonists: d-serine for GluN2A and glycine for GluN2B-containing receptors (*Papouin et al., 2012*). While our data show that GluN2B-containing NMDARs are selectively activated by ambient glutamate, they do not distinguish between potential synaptic and extra-synaptic locations of this receptor pool, in addition, we did not examine the role of triheteromeric NMDARs in regulation of protein synthesis and maintenance of excitatory synaptic strength. Regardless of their subcellular location, however, our data strongly implicate a role for NMDARs containing GluN2B in the rapid antidepressant actions of ketamine, via their ability to directly suppress mTOR signaling and limit protein synthesis in principal cortical neurons.

## Materials and methods

### Generation and genotyping of GluN2BΔCtx mice

GluN2B principal neuron KO mice were obtained by crossing GluN2B 'floxed' mice (*Brigman et al., 2010*) and NEX-Cre mice (*Goebbels et al., 2006*). For genotyping GluN2B floxed allele, primers AGG GTT TTA CAT ACC CCA GGC TGC and AGA GGA TCT ACC AGT AAC ATG C were used to produce a 412-bp WT fragment and 326-bp fragment from the floxed allele. NEX-Cre genotyping involved three primers: NEX.148 s (GAG TCC TGG AAT CAG TCT TTT TC), NEX.as (AGA ATG TGG AGT AGG GTG AC), and Cre.a (CCG CAT AAC CAG TGA AAC AG), producing a 770-bp fragment from WT NEX allele, 525-bp fragment from the NEX-Cre allele. GluN2B$^{flox/+}$ : NEX$^{Cre/+}$ mice were crossed to obtain GluN2B$^{flox/flox}$ : NEX$^{Cre/+}$ mice, which we refer to as GluN2BΔCtx (or 2BΔCtx). NEX$^{+/+}$ : GluN2B$^{flox/+}$ or NEX$^{+/+}$ : Glun2B$^{flox/flox}$ mice served as controls. Mice were housed on a 12/12 hr light–dark cycle, at ~20°C and ~55% humidity. Regular rodent chow and tap water were available *ad libitum*. All pups, including controls, were fed daily with Ensure starting at postnatal day 18 (P18) until P25 to increase survival rates of the mutants. Mice were weaned at P30 by gender. Juveniles were group-housed to not >5 animals/cage.

### Behavioral experiments

Behavioral experiments were performed on male mice aged P50–P70 and these same animals were used to generate all of the data in *Figure 3*. Sex was not determined for embryos used to generate cell cultures; however, all of the remaining data were generated from male mice. All protocols were approved by the Tulane University IACUC. Mice were brought to the test room and habituated for at least 1 hr before testing. Videos were taken using a Canon PowerShot A2200 camera. TST was performed as previously described (*El Yacoubi et al., 2003*). Mice were suspended using adhesive tape and video recorded for 6 min. Two mice were tested simultaneously with an opaque screen separating them. Videos were scored offline for immobility blind to genotypes and treatments. Forced swim test was carried out as described (*Porsolt et al., 1977*; *El Yacoubi et al., 2003*). Mice were introduced to a cylinder filled with room temperature water. The cylinder was 20 cm in diameter and filled about 12 cm deep to prevent mice using their tails to support themselves. Two mice were tested simultaneously with a screen to separate the cylinders. Videos were recorded from a top-mounted camera for 6 min. After the testing, mice were rubbed dry and placed under an infrared lamp for about 20 min. Immobility in the last 4 min of 6 min test session was scored blind to genotype and treatment. Open field test was conducted using Accuscan open field system. Mice were introduced into the arena at a fixed corner, and they were able to explore the arena freely. Ketamine and saline were administered via i.p. injection and ketamine concentration used was 50 mg/kg, a dose we empirically found to be necessary to elicit anti-despair-like behavior in our control animals. Consistent with other results (*Lindholm et al., 2012*), our data showed that the effectiveness of ketamine on despair-like behavior on P50–P70 mice required higher doses than reported for rats and adult mice at older ages. Corticosterone (Sigma–Aldrich, St. Louis, MO) was dissolved in ethanol at 5 mg/ml and further diluted in drinking water to 25 µg/ml. Water containing corticosterone or ethanol vehicle (0.5%, Veh) replaced animals' regular drinking

water and was available at all times in the home cage, providing the only source of hydration. All food provided was soaked in the same corticosterone containing water. Exposure began between P30 and P40. After continuous treatment for 20 days, mice were assessed in TST. The same mice were then used for mEPSC recordings. For in vivo injections, ceftriaxone was dissolved in saline and mice were injected for 20 days at 200 mg/kg as per previous reports (*Mineur et al., 2007*) and tested in the TST on day 21 which fell between P50 and P70 consistent with the ketamine experiments. NDGA was dissolved in ethanol to 100 mg/ml and then further diluted to 10 mg/ml in olive oil. Olive oil containing 10% ethanol was used as the vehicle control. P50–P70 mice were injected twice 12 hr apart with 0.1 ml of either NDGA or vehicle. 12 hr after the second injection animals were tested in the TST. In chronic variable stress paradigm P30–P50 group-housed male mice were stressed in a pseudo-random order for 3 weeks using the following: cold room (2 hr at 4°C); shaker, and no bedding (1 hr at 50–60 rmp followed by overnight in clean cage with no bedding); warm swim and single housing (30°C for 20 min followed by overnight in single-housed cage); cold swim (20°C for 10 min); wet bedding (2 hr with wet bedding).

## Tissue culture experiments

All tissue culture methods have been previously published (*Hall et al., 2007*; *Wang et al., 2011a*). To validate glutamate-reuptake modulating drugs' ability to alter levels of ambient glutamate, we pretreated cultured cortical cells (15–18 DIV) for 1 hr with 20 µM dl-TBOA (to suppress EAAT-mediated reuptake) or 4 µM NDGA (to enhance EAAT-mediated reuptake—Santa Cruz Biotechnology, Dallas, TX). To chronically alter the tonic current, we pre-treated cells for 7 days before recording. Under these conditions, NDGA caused widespread cell death, and so for these recordings, we substituted a different enhancer of EAAT function, ceftriaxone (100 µM–Sigma).

## Biochemical methods

For synaptoneurosome preparations (generated at P50–P70), we followed the methods of *Li et al. (2010)*. Epifluorescence-based imaging was performed per standard protocols, previously described in the reference *Hall et al. (2007)* and FUNCAT method was as per that described in the reference *Dieterich et al. (2010)*.

## Stereotaxic injections

Stereotaxic bilateral injections into the medial prefrontal cortex were performed on P40- to P50-day-old, homozygous, loxP-based conditional allele containing mice with 400 nl of AAV-CaMKIIa-GFP-Cre virus ($8 \times 10^{12}$ genome copy/ml). Mice were sacrificed for electrophysiological recordings in acute brain slices 10–14 days later, as detailed below.

## Acute brain slice recordings

For acute slice recordings juvenile (P15–P21, for ifenprodil sensitivity assessment) and adult (P50–P70, for all other experiments) control and 2BΔCtx mice were anesthetized with isoflurane and decapitated. Brains were removed and immediately placed into ice-cold ACSF containing high $Mg^{2+}$ (8 mM) and low $Ca^{2+}$ (0.25 mM) to promote slice health. 350-µm thick coronal slices from the PFC, defined as caudal to the olfactory bulb and rostral to the commissure of the corpus callosum were obtained using a Leica VT1200 vibratome. Slices were transferred to a holding chamber where they were incubated in bicarbonate buffered ACSF at room temperature for at least 45 min before transferring to a recording chamber for whole-cell voltage clamp recording. ACSF solutions were bubbled with 95%$O_2$/5%$CO_2$ at all times to maintain consistent oxygenation and pH. Synaptic activity was recorded from acute brain slices while perfused at room temperature in ACSF. Borosilicate glass pipettes were filled with a cesium-substituted intracellular solution as previously described (*Hall et al., 2007*). Pipette resistances ranged 4–7 MΩ. Series access resistance ranged from 7 to 15 MΩ and was monitored for consistency. Recordings were discarded if leak current rose above 300 pA. In slice recordings, layer II/III pyramidal neurons were targeted for recording. Recorded cells were confined to medial cortex in the more caudal slices (to target PrL and IL cortex) and were medial and dorsal in more rostral slices, defined by appearance of forceps minor corpus callosum, observable in live differential interference contrast images. Synaptic responses were evoked using concentric bipolar-stimulating electrodes placed approximately halfway across the cortical layers in coronal orientation to evoke stimulation of intracolumnar axons. Current decay tau values were determined using a single exponential decay function in IgorPro: $y = y_0 + a^{(-(x - x0/tau))}$ and fitting between the current peak and Δ50 ms.

## Acknowledgements

This work was supported by a grant from the National Institute for Mental Health (MH099378-01 to BJH) and a NARSAD Young Investigator Award from the Brain and Behavior Research Foundation (YIA18996 to BJH). OHM was supported by a Louisiana Board of Regents Fellowship. We thank Klaus Nave for the NEX-Cre mice, Eric Delpire for the GluN2B conditional KO mice, and Gary Westbrook and Masayoshi Mishina for the GluN2B KO mice. GluN2B conditional KO mice were generated by the NIH supported Integrated Neuroscience Initiative on Alcoholism (AA13514).

## Additional information

### Funding

| Funder | Grant reference number | Author |
|---|---|---|
| National Institute of Mental Health | MH099378-01 | Benjamin J Hall |
| Brain and Behavior Research Foundation | YIA18996 | Benjamin J Hall |
| Louisiana Board of Regents | Graduate Fellowship | Oliver H Miller |

The funders had no role in study design, data collection and interpretation, or the decision to submit the work for publication.

### Author contributions

OHM, LY, Conception and design, Acquisition of data, Analysis and interpretation of data, Drafting or revising the article; C-CW, EAH, YZ, Acquisition of data, Analysis and interpretation of data; ED, Made a critical contribution by generating and making available the conditional GluN2B knockout animals, Contributed unpublished essential data or reagents; BJH, Conception and design, Analysis and interpretation of data, Drafting or revising the article

### Ethics

Animal experimentation: This study was performed in strict accordance with the recommendations in the Guide for the Care and Use of Laboratory Animals of the National Institutes of Health. All of the animals were handled according to approved institutional animal care and use committee (IACUC) protocols of Tulane University (#0363R and 0364R).

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
