## [Decision Letter]

Thank you for sending your work entitled “GluN2B-Containing NMDA Receptors Regulate Behavioral Despair and the Rapid Antidepressant Actions of Ketamine” for consideration at *eLife*. Your article has been favorably evaluated by a Senior editor and 2 reviewers, as well as Gary Westbrook (Reviewing editor). The Reviewing editor and the reviewers discussed their comments before we reached this decision, and the Reviewing editor has assembled the following comments to help you prepare a revised submission.

The reviewers thought these were interesting studies investigating the role of GluN2B subunit in mediating depression-related behaviors. The authors report that in vivo deletion of GluN2B from principal cortical neurons mimics and occludes the action of ketamine on depression-related behaviors and on excitatory synaptic transmission. Ketamine-induced increases in mTOR activation in synaptic protein synthesis also were mimicked and occluded in the GluN2B mutant mice. The experiments in the last section of the paper attempt to show that GluN2B-containing NMDA receptors are uniquely activated by ambient glutamate to regulate excitatory synaptic transmission. Reviewer 1 was positive about the paper: “This is an outstanding paper. It is well written, thorough, and convincing illustrates the critical role of GluN2B-containing NMDA receptors in the fast-acting antidepressant properties of ketamine. This paper combines genetic, pharmacological, electrophysiological, and behavioral techniques to demonstrate that ketamine, an NMDA antagonist, acts to suppress GluN2B-containing NMDA receptors to promote protein synthesis and produces an antidepressant behavioral response.” Reviewer 1 had two concerns (#1 and #2) below. Reviewer 2, although positive about the paper, had more serious concerns about the conclusions regarding ambient glutamate and the manipulation of transporter currents (see #3 and #4) below. We anticipate that you will either need to add some additional experiments to address the issues and/or modify your conclusions

1) The authors use mTOR phosphorylation as a measure for upregulated mTOR signaling. Although often correlated with mTOR activation, phosphorylation of mTOR is not always indicative of mTOR complex 1 (mTORC1) signaling that is coupled to translational control. Moreover, this measure cannot distinguish between mTORC1 and mTORC2, which phosphorylates Akt. Thus, the authors should determine whether ketamine (or removal of GluN2B) can trigger increases in the phosphorylation of the mTORC1 substrates 4E-BP and S6K1 (or their downstream effectors, which are eIF4E-eIF4G interactions and S6 phosphorylation, respectively).

2) The authors should show that inhibition of mTORC1 with rapamycin is able to block the enhanced protein synthesis observed in the GluN2BΔCtx neurons using the FUNCAT method. A positive result for this experiment would provide more conclusive evidence that the biochemical and behavioral alterations observed in these mice are due to increased mTORC1 activation and not through an alternate signaling pathway.

3) The lack of tonic NMDAR current in recordings from 2BdeltaCtx slices is somewhat perplexing. GluN2B containing NMDARs have a higher affinity for glutamate than 2A receptors but the EC50 is only ∼2.5 fold lower for the former (Hansen et al., 2014; Neuron 81:1084). In addition, the probability of opening of 2A receptors is higher than that of 2B receptors (Erreger et al., 2005; J. Physiol. 563:345), which would somewhat make up for the lower affinity. As the authors point out, 2A receptors also have a lower affinity for the coagonist. It would seem to be important to test whether there is any tonic current when either ambient glutamate or ambient glycine/D-serine is elevated either with TBOA or added D-serine to slices from 2BdeltaCtx mice and the 2B to 2A mice, or is it merely because these animals do not express enough receptors of any type. In this regard, is the effect of activating 2B receptors because of Ca2+ influx or is it some direct, metabotropic activation via 2B receptors? If it is the former, enhancing 2A activation, should have the effects that the authors observe with 2B activation.

4) The other concern is the use of nordihydroguaiaretic acid and ceftriaxone. Neither has been shown to have a functional effect on astrocyte transporter currents in slices. The Rothstein et al. Nature paper that is cited in the manuscript tested ceftriaxone on synaptically-evoked currents in astrocytes in hippocampus but there was not a significant effect. Similarly, norhydroguaiaretic has not been tested on transporter currents in intact preparations. Finally, both of these drugs have many actions (PubMed search) and, while the present manuscript reports effects in vivo that could be the result of modification to transport and ambient levels of glutamate, this is correlative, not necessarily causal, and thus not terribly convincing.

---

## [Author Response]

*1) The authors use mTOR phosphorylation as a measure for upregulated mTOR signaling. Although often correlated with mTOR activation, phosphorylation of mTOR is not always indicative of mTOR complex 1 (mTORC1) signaling that is coupled to translational control. Moreover, this measure cannot distinguish between mTORC1 and mTORC2, which phosphorylates Akt. Thus, the authors should determine whether ketamine (or removal of GluN2B)* can *trigger increases in the phosphorylation of the mTORC1 substrates 4E-BP and S6K1 (or their downstream effectors, which are eIF4E-eIF4G interactions and S6 phosphorylation, respectively)*.

We have added data demonstrating that phosphorylated p70S6 kinase (p-p70S6K) is elevated in 2BΔctx pre-frontal cortex of P50-P70 animals, supporting our FUNCAT data showing enhancement of mTOR-dependent protein synthesis in this brain region in the absence of GluN2B, see Figure 6 below and added text.Author response image 1.Western blot data from P50-P70 PFC showing increased p-P70S6K signal in lysates from 2BdeltaCTX animals (left) and 2B->2A animals (right).

Additionally we include protein quantification from cortex of transgenic mice whose GluN2B subunit has been replaced with GluN2A (2B◊2A mouse). In these animals total NMDAR contribution is recovered in the absence of GluN2B (Wang et al., 2011; Neuron 72:789-805). In 2B->2A samples we also observed elevated levels of p-p70S6K suggesting that it is the loss of GluN2B specifically, rather than a general loss of NMDAR signaling that is responsible for elevated mTORC1 activation (see below). Finally, we also refer to the original reference characterizing these 2B->2A mice in which elevated levels of p-p70 s6K was revealed in the absence of GluN2B in cortical dendrites of 2A◊2B mice by immunofluorescence, Figure 6 from Wang et al., 2011 (this figure shows increased p-P70S6K signal in 2B null neurons by immunofluorescence. In these 2B->2A cells the GluN2B genetic locus has been replaced with 2A cDNA. In these cells total NMDAR density is comparable with WT neurons but the receptor population is completely 2A-containing).

*2) The authors should show that inhibition of mTORC1 with rapamycin is able to block the enhanced protein synthesis observed in the GluN2BΔCtx neurons using the FUNCAT method. A positive result for this experiment would provide more conclusive evidence that the biochemical and behavioral alterations observed in these mice are due to increased mTORC1 activation and not through an alternate signaling pathway*.

We have added data (see also Figure 4 ) demonstrating that the increase in protein synthesis observed after removal of GluN2B from cortical principle neurons is rapamycin- sensitive, suggesting that mTORC1-dependant translation is responsible.

*3) The lack of tonic NMDAR current in recordings from 2BdeltaCtx slices is somewhat perplexing. GluN2B containing NMDARs have a higher affinity for glutamate than 2A receptors but the EC50 is only ∼2.5 fold lower for the former (Hansen et al., 2014; Neuron 81:1084). In addition, the probability of opening of 2A receptors is higher than that of 2B receptors (Erreger et al., 2005; J. Physiol. 563:345), which would somewhat make up for the lower affinity. As the authors point out, 2A receptors also have a lower affinity for the coagonist. It would seem to be important to test whether there is any tonic current when either ambient glutamate or ambient glycine/D-serine is elevated either with TBOA or added D-serine to slices from 2BdeltaCtx mice and the 2B to 2A mice, or is it merely because these animals do not express enough receptors of any type. In this regard, is the effect of activating 2B receptors because of Ca2+ influx or is it some direct, metabotropic activation* via *2B receptors? If it is the former, enhancing 2A activation, should have the effects that the authors observe with 2B activation*.

This is a very interesting question. To address this we did additional experiments. We used in vivo viral injection of CRE:GFP in 2B homozygous floxed animals to generate null neurons in a mosaic fashion. In acute brain slices from these animals we confirmed that 0Mg^2+^ treatment does not result in increased RMS signal in 2B null (cre-expressing) neurons, consistent with our previous data. However, additional bath application of 25 micromolar NMDA in the absence of Mg^2+^ did activate a tonic current in these neurons, evident as a strong increase in holding current and increased RMS signal (see Figure #3 below). This shows indeed that GluN2A-containing NMDARs can be activated by ambient glutamate and the increased RMS signal is likely due to current flux. However, they also substantiate the hypothesis that non 2B-containing NMDRs are much more sensitive to ambient agonist. We believe the difference in sensitivity could reflect differential access to agonist due to unique subcellular localizations of the receptor subtypes and/or the presence of interacting proteins in neurons that modify sensitivity of the receptor complexes. It is notable that characteristic properties of receptor physiology are different when comparing receptors in neurons with those transiently expressed in heterologous systems. For example, while 2A-containing receptors exhibit greater open channel probability in heterologous systems, measuring the progression of activity-dependent channel block does not support the same conclusion in neurons but rather data in neurons show evidence for similar open probability kinetics (Wang et al., 2011; Speed and Dobrunz, 2009; Chavis and Westbrook, 2001). We have added a description of the experiment shown here in Figure 7 to the Results and Discussion sections.Author response image 2.This figure shows GFP:cre expressing, 2B null neurons in acute PFC slices (top). Challenge of these slices with 0Mg2+ did not initiate a tonic current however the NMDAR population could still be activated by bath application of NMDA.

*4) The other concern is the use of nordihydroguaiaretic acid and ceftriaxone. Neither has been shown to have a functional effect on astrocyte transporter currents in slices. The Rothstein et al. Nature paper that is cited in the manuscript tested ceftriaxone on synaptically-evoked currents in astrocytes in hippocampus but there was not a significant effect. Similarly, norhydroguaiaretic has not been tested on transporter currents in intact preparations*.

As the reviewer points out, both ceftriaxone and NDGA have other effects (beta-lactam antibiotic and lipoxygenase inhibitor, respectively), and neither drug’s role has been directly tested on astrocyte transporter currents. However, the literature demonstrably supports a role for both drugs in enhancing EAAT-mediated glutamate reuptake. [63] demonstrate that ceftriaxone activates the GLT1(EAAT2) promoter in cultured human astrocytes (Figure 2 in [63]), induces protein expression of EAAT2 in rodent spinal cord cultures (Figure 1) and increases glutamate transport in cortical tissue from drug-treated mice (Figure 3). [60] show that repeated ceftriaxone treatment results in a reduction of extracellular glutamate in the intact brain (Figure 1 in [60]), likely due to this enhancement of EAAT2 expression.

In fact, [62] show that knock-down of EAAT2 results in decreased transport function and increased ambient glutamate (Figure 2 in [62]), this citation has been added to the manuscript. Additionally, Guo et al., 2003 show that over expression of EAAT2 results in enhanced reuptake in cortex (Figure 3 in [23]), this citation has also been added to the manuscript. Finally, Boston–Howes et al., 2008 demonstrate that NDGA treatment results in enhanced DHK (selective EAAT2 inhibitor)-reversible enhancement of glutamate uptake in MN-1 cells, suggesting that NDGA causes upregulation of EAAT2-mediated transport (Figure 2 in [9]) and also demonstrate that NDGA-treated intact rodents show enhanced glutamate reuptake in spinal cord preparations (Figure 4 in [9]).

*Finally, both of these drugs have many actions (PubMed search) and, while the present manuscript reports effects* in vivo *that could be the result of modification to transport and ambient levels of glutamate, this is correlative, not necessarily causal, and thus not terribly convincing*.

It is of course possible that the two drugs work through two separate mechanisms that converge to alter tonic 2B activation and cause the reduction in depression-like behavior we observe. However, since ceftriaxone and NDGA have no published common targets (other than in enhancing EAAT2-mediated glutamate uptake), the most likely explanation we have at this time is that they are both acting through a common pathway to enhance tonic GluN2B activation and cause the resultant downstream effects we observe. This is further evidenced by the lack of any effect of NDGA on the 2BΔCtx mice, both in mini EPSC frequency and behavior, where tonic 2B activation is not possible. More targeted enhancers of EAAT2 would be ideal but are not currently commercially available. In light of this we have kept our original interpretation but we now also mention the caveat of off-target effects in the manuscript text.